



# Dust record in an ice core from tropical Andes (Nevado Illimani – Bolivia), potential for climate variability analyses in the Amazon basin

Filipe G. L. Lindau[1], Jefferson C. Simões[1,2], Rafael R. Ribeiro[1], Patrick Ginot[3], Barbara Delmonte[4], Giovanni Baccolo[4], Stanislav Kutuzov[5], Valter Maggi[4], and Edson Ramirez[6]

[1]Centro Polar e Climático, Universidade Federal do Rio Grande do Sul, Porto Alegre, 91501-970, Brazil
[2]Climate Change Institute, University of Maine, Orono, ME 04469, USA
[3]Univ. Grenoble Alpes, CNRS, IRD, Grenoble INP, IGE, 38000 Grenoble, France
[4]Environmental and Earth Sciences Department, University Milano-Bicocca, Milan, 20126, Italy
[5]Institute of Geography, Russian Academy of Sciences, Moscow, 119017, Russia
[6]Instituto de Hidráulica e Hidrología, Universidad Mayor de San Andrés, La Paz, Bolivia

**Correspondence:** Filipe G. L. Lindau (filipe.lindau@outlook.com)

**Abstract.** Understanding the mechanisms controlling glacial retreat in the tropical Andes can strengthen future predictions of ice cover in the region. As glaciers are a dominant freshwater source in these regions, accurate ice cover predictions are necessary for developing effective strategies to protect future water resources. In this study, we investigated a 97-year dust record from two Nevado Illimani ice cores to determine the dominant factors controlling particle concentration and size distri-

bution. In addition, we measured the area of a Nevado Illimani glacier (glacier n°8) using aerial photographs from 1956 and 2009. We identified two dustier periods during the 20th century (1930s–1940s and 1980s–2016), which were linked to reduced moisture transport from the Amazon basin. This promoted an unprecedented increase in the percentage of coarse dust particles (CPPn, $\varnothing > 10$ µm) during the 1990s, as drier local conditions favored the emission and deposition of coarse particles on the glacier. Moisture advection from the Amazon basin to Nevado Illimani was influenced by tropical North Atlantic sea surface

temperatures (TNA), which was supported by the correlation between TNA and CPPn ($r = 0.52$). Furthermore, glacial retreat has been accelerating since the 1980s, and a notable relationship between CPPn and the freezing level height (FLH, $r = 0.41$) was observed. This suggests that higher FLHs promote glacial retreat, which exposes fresh glacial sediments and facilitates the transport of coarse dust particles to the Nevado Illimani summit. Therefore, both the area of glacier n°8 and the ice core record of coarse dust particles were found to respond to climate variability—particularly to the warmer conditions across the southern

tropical Andes and drier conditions over the Amazon basin.

## 1 Introduction

The tropical Andes has undergone pronounced climatic changes over the past few decades. Tropospheric warming has significantly accelerated glacial retreat, resulting in the fastest rates of retreat since the mid-Little Ice Age (mid-seventeenth to early eighteenth century) (Rabatel et al., 2013). Changes in glacier mass, length, area, and volume have been recorded since





the 16th century and are strong indicators of climate change (Bojinski et al., 2014; Zemp et al., 2015). However, quantitative information on the climate-glacier relationship requires a multi-proxy approach that combines glacier geometry data with other paleoclimatic proxies, such as ice core records (Solomina et al., 2007; Jomelli et al., 2009).

The presence of large dust particles ($> 8$ μm diameter) in a Renland ice core (RECAP, 71.30°N, 26.72°W, 2315 m above sea level, asl) was linked to the activity of local dust sources and inferred past changes in the ice sheet margin at the east coast

of Greenland (Simonsen et al., 2019). The RECAP dust record was dominated by coarse particles during the Holocene and previous interglacial period, inferring the retreat of the East Greenland ice sheet margin and the exposure of sediment-rich areas, which increased the supply of coarse material. Similarly, the higher snowline elevation during the Holocene compared with the Last Glacial Maximum (LGM, approximately 20 ka BP) in the southern tropical Andes had caused an 8-fold increase in dust concentrations in the Sajama ice core (18.1°S, 68.88°W, 6542 m asl) (Thompson et al., 1998). Moreover, according to

Nevado Illimani dust records (hereafter Illimani, 16.62°S, 67.77°W, 6438 m asl), Holocene dust concentrations were 2.5 times higher than that of glacial periods due to the drier local conditions (Ramirez et al., 2003). Higher dust concentrations during warm periods emphasizes the importance of local dust sources, as the dust cycle generally decreases during warm periods on the global scale (Maher et al., 2010). However, the Sajama and Illimani studies only assessed the total mineral concentration and did not consider the variability of mineral particle size during these climatic transitions. Due to the close link between

coarse atmospheric dust and climatic and environmental processes (Baccolo et al., 2018; Simonsen et al., 2019; Lindau et al., 2020), it is necessary to investigate the relationship between coarse dust particles ($> 10$ μm diameter), glacial evolution, and climate variability in the southern tropical Andes.

This study aimed to assess the relationship between ice core dust records and current glacial retreat in the tropical Andes. We analyzed the dust concentrations and particle sizes in firn and ice samples from Illimani covering 1919–2016 AD to assess

past dust aerosol variability. The samples were collected during two drilling campaigns in 1999 and 2017. To better understand the causes of dust variability, our results were compared to other climate records, meteorological observations, and aerial photographs of an Illimani glacier covering the period 1956–2009.

The Illimani is located in Cordillera Real, one of the four mountain chains forming the Eastern Cordillera of the Bolivian Andes (Fig. 1a). This mountain is 50 km southeast of the Bolivian capital La Paz and 180 km southeast of Lake Titicaca. The

precipitation is distributed over 9 months of the year (Vimeux et al., 2009), with a clear dry (April to August, austral winter) and wet season (November to March, austral summer). An abrupt transition between the two seasons occurs between March and April, and intermediate conditions occur in September and October (Vimeux et al., 2005). Dust from the Illimani was almost exclusively composed of soil-derived particles during both summer and winter throughout the 20th century (Correia et al., 2003). Mineralogy, rare earth elements, and strontium/neodymium isotope ratios can also determine the local and re-

gional (southern Altiplano) provenances of dust particles (Delmonte et al., 2010; Lindau et al., 2020).



## 2 Material and Method

### 2.1 Field campaigns and sampling

A joint expedition of scientists from the French Institut de Recherche pour le Developpement (IRD) and the Paul Scherrer
Institute (PSI) recovered two deep ice cores (depths of 136.7 and 138.7 m) from Illimani in June 1999 at an altitude of 6350
m asl (Fig. 1b). Both ice cores were drilled with a Fast-Electromechanical Lightweight Ice Coring System (FELICS) (Ginot
et al., 2002) producing sections of up to 0.9 m in length and 7.8 cm in diameter (Knüsel et al., 2003). A 23.8 m firn core was
recovered from approximately the same drilling site in June 2017 using an EM-100-1000 electromechanical drill (Cryosphere
Research Solutions, Columbus, Ohio, USA), which produced a core of 10 cm in diameter. The 2017 expedition was composed
of a French, Russian, Bolivian, and Brazilian team as part of the Ice Memory project (https://www.ice-memory.org), which
extracted two additional cores down the bedrock (136 and 134 m).

The upper 45 m (35 m in water equivalent, w.e.) of one of the 1999 ice cores (IL1999) were cut into sections of approximately
10 cm (from the top to 40 m) and 7 cm (from 40 to 45 m) at the Institut des Géosciences de l'Environnement (IGE, University
Grenoble Alps, France), providing a total of 666 samples (Ramirez et al., 2003). The firn core recovered in 2017 (IL2017)
was cut into sections of approximately 5 cm at the EuroCold (University of Milano-Bicocca, Italy), providing 464 additional
samples.

### 2.2 Dust concentration and size distribution

The dust content in IL1999 was analyzed in 2002 at IGE using a Beckman Coulter Multisizer IIe with a 50 µm orifice. The
instrument counted the number of particles with spherical equivalent diameters of 0.67–20.89 µm, which was divided into 256
size intervals. The IL2017 dust content was measured in 2018 at the EuroCold (University Milano-Bicocca) using a Beckman
Coulter Multisizer 4 equipped with a 100 µm orifice. The instrument measured the particle distribution from 2 to 60 µm,
which was divided into 400 size intervals. To increase sample conductivity for the Coulter Counter analysis, a clean prefiltered
NaCl solution was added to samples to obtain a final NaCl concentration of approximately 1% (mass fraction). To reduce the
sedimentation of large particles, all samples were shaken before measurement. Two or three measurements were conducted on
each sample depending on the total volume, and each measurement consumed 0.5 mL of sample. The mean relative standard
deviation between the measurements was approximately 10% for both IL1999 and IL2017 for the 2–20 µm particle size range.
Blank background counts represented less than 0.5% of the mean particle concentration in the samples.

In order to compare the IL1999 and IL2017 analyses, we calculated the particle number concentration for the size range mea-
sured for both cores (2–20 µm). The coarse particle percentage in terms of number (CPPn) is defined as the ratio between the
number of particles of > 10 µm diameter and the total number of particles within the 2–20 µm size range. To assess the sim-
ilarities between the two ice cores, we used an additional core (IL2009) spanning the period 1993–2009 (dated by multiproxy
annual layer counting). The dust concentration for IL2009 was measured using an Abakus laser particle sensor (measuring
particles in the range of 1–100 µm at 1 cm intervals) connected to an ice core melting continuous flow system. The overlapping





period between the integrated IL1999 + IL2017 Coulter Counter dust record and the IL2009 Abakus dust concentration record
      suggests good reproducibility (Fig. S1).

      The dust record in Illimani was then compared to the one in Quelccaya Ice Cap (13.93°S, 70.83°W, 5670 m asl) (Thompson
      et al., 2013). The annually resolved dust concentration record from the Quelccaya ice core, covering the 1919–2003 period,
      was obtained from the World Data Center for Paleoclimatology (NCDC/NOAA) available on https://www.ncdc.noaa.gov/data-
access/paleoclimatology-data/datasets/ice-core.

### 2.3    Chronology and snow accumulation rates

      IL2017 was dated via annual layer counting (ALC) based on its well-preserved seasonal dust content, $Ca^{2+}$, and $\delta$D; the age of
      the core ranged from 1999 to 2016 (Lindau et al., 2020). For IL1999, the reference horizons of well-known volcanic eruptions
(Pinatubo, 1991; El Chichón, 1982; Agung, 1963) were determined by the electrical conductivity method (ECM) and by the
      content of fluoride, chloride, and sulfate (De Angelis et al., 2003). In addition, a tritium peak was attributed to the year 1964,
      signifying the nuclear weapons testing during 1964–1967 (Knüsel et al., 2003). The resulting chronology for IL1999 was 1919
      to 1999 (Fig. 2), with an estimated uncertainty of ± 2 years from 1919 to 1941 and ± 1 year for 1941–1999 (De Angelis et al.,
      2003). As a similar approach was used to date IL2017, we estimate an uncertainty of ± 1 year for the period 1999–2016.
The annual snow accumulation rates were reconstructed from the annual layer thicknesses, considered as the depth (in w.e.)
      between winter dust peaks. However, the thickness of a layer deposited on the surface decreases with depth due to compression
      and stretching. In order to establish the original thickness of the annual layers an exponential fit regression was applied to the
      layer thickness with the depth (Nye, 1963). This regression equation was used to calculate the accumulation following Eq. (1):

$$acc = \frac{(p_i - p_{i-1})a^{-b}}{a^{-bp_i}} \tag{1}$$

      where acc is accumulation, $p_i$ is depth, $(p_i - p_{i-1})$ is the layer thickness, and a and b are the constants from the best exponential
      fit regression. It is important to note that this method focuses on the interannual variability and can therefore mask the overall
      trend across the entire time range (Vimeux et al., 2009).

### 2.4    Glacier and climatic data

      In this study, we used twelve aerial photographs of the n°8 glacier on the Illimani (Fig. 1b) during the dry seasons of 1956,
      1963, 1975, 1983, and 2009 (available from the Bolivian Servicio Nacional de Aerofotogrametria [SNA]). We chose this glacier
      because it showed the best representation in the oldest available air photographs for this area. It is also a relatively large glacier
      (1.95 km² in 2009) in Bolivia, and most of its accumulation area is above 5000 m asl.
The photographs were digitized with a resolution of 14 μm using a photogrammetric scanner. The internal and external ori-
      entation was performed through a digital photogrammetric workstation equipped with a Planar 3D stereoscopic visualization





system and the Leica Photogrammetry Suite (LPS) digital system. For the camera calibration, the radial distortion of the lenses, the focal length and the position of the fiducial marks were considered (Table S1). Eight control points were produced during fieldwork in 2010 using a pair of Astech Zmax L2 differential GPS. The ice-covered areas and drainage basins were digitized

using a UTM-WGS 84 (zone 19 south) reference system. The glacier terminus was manually determined based on the aerial photographs; subsequently, we extracted a terrain digital elevation model (DEM) for each studied year. Volume glacier changes were then calculated by subtracting the DEMs of the different years (Ribeiro et al., 2013).

Freezing level heights (FLH) for the Illimani were calculated using monthly temperatures and geopotential heights from the NCEP/NCAR reanalysis (https://psl.noaa.gov/data/gridded/data.ncep.reanalysis.html) for the period 1948–2017. The compar-

ison of FLH with the elevation from the glacier front helped to explain the strong ablation rates in Bolivian glaciers and their consequent retreat (Rabatel et al., 2013). The data were centered at 17.5°S, 67.5°W (2.5° × 2.5° resolution), and the levels around the FLH (500 and 600 hPa) were examined for a transition from positive to negative temperature. The FLHs were then linearly interpolated from the geopotential heights of the transition levels (similar to Bradley et al., 2009) and annually resampled.

Precipitation records at annual resolution for Apolo, Rurrenabaque, and Sapecho (Fig. 1a) were provided by the SENHAMI (Servicio Nacional de Meteorologia e Hidrologia, Bolivia) network (www.senhami.gob.bo/sismet). We used the Extended Reconstructed Sea Surface Temperature (ERSST, version 5, Huang et al., 2017) for the tropical Pacific (Niño 4 region, 5°N–5°S, 160°E–150°W) and tropical North Atlantic (TNA, 20°N–5°N, 60°W–30°W). We also employed the reanalysis dataset from the European Center for Medium Range Weather Forecasts (ECMWF, ERA5). All of the data were obtained from the KMNI

Climate Explorer (http://climexp.knmi.nl/getindices) at annual resolution.

## 3  Results and discussion

### 3.1  A 97-year tropical Andean dust record

The mean annual dust concentration in Illimani for the period 1999–2016 was 13,189 part mL$^{-1}$. We compared this new com-

posite Illimani record with the Quelccaya ice core record (Thompson et al., 2013) and identified two periods of relatively high dust concentrations at both sites (Fig. 3a): 1) mid-1930s–mid-1940s, and 2) mid-1980s–2016 (the Quelccaya record finishes in 2002). In agreement, reconstructed annual precipitation from tree-ring growth in western Altiplano during 1300–2006 AD inferred a severe drought event between 1930 and 1948 (1940 was the second driest year in the entire record), which was preceded by wet conditions since the 19th century (Morales et al., 2012). In addition, the reconstructed snow accumulation in

Illimani showed the lowest rate during this period (0.35 m w.e. a$^{-1}$ in 1937), and the snow accumulation rates at Quelccaya were also relatively low (Fig. 3b). Furthermore, three of the most extreme dry years in Altiplano had occurred after the 1980s (1982, 1994, and 2006; Morales et al., 2012). We also observed a shift to lower snow accumulation rates in the mid-1980s for both Illimani and Quelccaya (Fig. 3b); this was attributed to lower moisture advection from the Amazon basin (Ribeiro et al., 2018), as Atlantic Ocean moisture via the Amazon basin is the primary moisture source for southern tropical Andes glaciers





(Hoffmann et al., 2003; Thompson et al., 2013).

We detected an unprecedented rise in the ratio of coarse particles (CPPn) at Illimani during the late 20th century– particularly since the 1990s (Fig. 3c). The mean CPPn over the period 1919–2016 was 1.5%, and annual values reached $> 3\%$ by the end of the 1990s. Figure 4 shows that the late 20th century increase in CPPn correlated ($p < 0.05$) with the decrease in specific humidity at the 500 hPa (5500 m height on average) level over the Bolivian Amazon. This suggests lower regional precipita-

tion and reduced moisture transport from the Amazon basin to the southern tropical Andes (Segura et al., 2020). Figure 4 also indicates reduced deep convection over the Bolivian Amazon (Espinoza et al., 2019). In agreement, CPPn showed a negative correlation with precipitation records from nearby meteorological stations (Fig. S2).

Moreover, CPPn variability was not directly linked to the percentage of giant dust particles ($\varnothing > 20\,\mu m$) in IL2017 (Lindau et al., 2020), as giant particles respond to deep convection during wetter periods in Altiplano under enhanced atmospheric

turbulence. Conversely, it is likely that coarse particles were more efficiently emitted and less efficiently scavenged under regional/local drier conditions—especially since the 1990s.

### 3.2   Large scale controls on the percentage of coarse dust particles

The late 20th century CPPn increase is consistent with the significant decline in precipitation over the Bolivian/ southern

Amazon basin during 1982–2017, which was attributed to a weakening moisture transport from the tropical North Atlantic Ocean (Espinoza et al., 2019). Warmer sea surface temperatures (SSTs) in the tropical North Atlantic (expressed by the TNA index) over the last decades (Fig. 5) altered the migration of the Intertropical Convergence Zone (ITCZ) toward warmer SSTs (Yoon and Zeng, 2010). As a result, the ITCZ was anomalously displaced northward under a warmer TNA; this weakened the northeasterly Atlantic trade winds and limited the moisture transport to the southern Amazon basin, leading to lower rainfall

(Yoon and Zeng, 2010; Marengo and Espinoza, 2016). In agreement, TNA was positively correlated with the 3-year moving average of CPPn ($r = 0.52$, $p < 0.01$), and both showed increasing trends since the 1990s (Fig. 5).

We also observed a weaker positive correlation between CPPn and SSTs in the tropical Pacific by comparing the grain size data with the El Niño 4 (5°N–5°S, 160°E–150°W) index ($r = 0.32$, $p < 0.05$, Fig. S3). Tropical Pacific SST anomalies are associated with anomalous upper-level westerlies over the southern tropical Andes, which inhibit convection and precipitation over

this region (Garreaud, 1999). Such anomalies, however, only explain approximately 13% of the total annual rainfall variability over the entire Amazon basin (Marengo, 1992; Espinoza et al., 2009).

### 3.3   Glacier area variability in Illimani

The glacial area of n°8 on Illimani reduced by 17% from 1956 to 2009 (Fig. 6), with an estimated area of 2.36 km$^2$ in 1983 and

1.95 km$^2$ in 2009. Fig. 6 indicates no significant variations in the glacial area of n°8 during 1956–1983, which suggests that the glacier was relatively stable. In addition, the DEMs for 1956, 1963, and 1975 showed high mean absolute errors (average of 6.3 m). From 1983 to 2009, we estimated a height reduction of $-13$ m and a negative mass balance of $-11.8 \pm 2.29$ m w.e.





This is in accordance with the estimated negative mass balance of the Zongo glacier (16.08°S, 68.28°W, Fig. 1a) for the same period ($-12$ m w.e.). In addition, this glacier lost 14.4% of its area from 1956 to 2006 and has been rapidly shrinking since 1975 (Soruco et al., 2009).

Bolivian glaciers experienced rapid retreat during the early 1980s. The Chacaltaya glacier (16.33°S, 68.12°W) showed a moderate mass-balance deficit during 1940–1963, and the period 1983–1998 was marked by drastic glacial shrinkage (Ramírez et al., 2001). Moreover, Dussaillant et al. (2019) estimated a strong negative mass balance of $-0.42 \pm 0.24$ m w.e. $\mathrm{yr}^{-1}$ for glaciers of the southern tropical Andes between 2000 and 2018. The Working Group on Snow and Ice of the International Hydrological Program (GTHN-PHI-UNESCO) estimated a 50% reduction in Bolivia's glacial surface from the 1970s to 2017, and Illimani lost 9.5 $\mathrm{km}^2$ of glacial coverage during the same period (Ribeiro et al., 2013). The timing of this accelerated retreat was attributed to tropospheric warming over the tropical Andes and a higher frequency of El Niño events, including changes in its spatial occurrence (Francou et al., 2003). The influence of warming in the tropical North Atlantic should also be considered, as warmer SSTs in 2005 had caused a strong negative mass balance of the Zongo glacier (Ribeiro et al., 2018).

### 3.4 Relationship between glacier retreat and the percentage of coarse dust particles

The timing of glacier n°8's retreat corresponds to the increase in coarse particle abundance in ice. We estimated an acceleration in the rising CPPn trend during 1987 (Fig. 7a). This period was obtained by fitting a continuous piecewise linear function to the 3-year averaged CPPn. Three line segments were obtained ($r^2 = 0.72$), and the slopes for 1920–1947, 1947–1987, and 1987–2015 were $-0.01$, 0.02, and 0.03 % $\mathrm{yr}^{-1}$, respectively. This infers a continued increase in dust sources to Illimani in recent years, which is likely caused by the increased exposure of soil sediments in deglaciated areas and a higher deposition of locally sourced coarse particles.

The accelerated retreat of glacier n°8 and the higher deposition of coarse dust particles at Illimani coincided with rising atmospheric temperatures during the late 20th century. Gilbert et al. (2010) quantified two warming phases based on englacial temperature measurements in a borehole at the Illimani drilling site during the 1999 campaign: 1) $+0.5$ °C from ~1920s to 1960, and 2) $+0.6$ °C from 1985 to 1999. This is in agreement with the relative areal stability of n°8 during 1956–1983. However, CPPn increased during the same period, which can be attributed to a larger retreat of the smaller Illimani glaciers (area $< 1$ $\mathrm{km}^2$) until the 1980s. The accumulation areas on the smaller glaciers are at lower elevations, and their entire surface can occasionally become an ablation zone. Moreover, the mean area loss of the small Illimani glaciers from 1963 to 1983 was 40% (a total loss of 1.6 $\mathrm{km}^2$) compared to a mean reduction of 8% for glaciers larger than 1 $\mathrm{km}^2$ (Ribeiro et al., 2013). Furthermore, the majority of glacier n°8 accumulation areas are above 5000 m asl.

We observed an increase in the FLH at Illimani (Fig. 7b) since 1950, which correlated with the CPPn trend ($r = 0.41$) at the 95% level. This is in accordance with the estimated 27.1 m per decade rise in freezing altitude over Cordillera Real from 1955 to 2011 (Rabatel et al., 2013). The FLH record in Fig. 7b also corresponds to the increased elevation of the glacier terminus; the mean terminus elevation of the largest Illimani glaciers was approximately 4770 m asl in 1963 and 4900 m asl in 2009 (Ribeiro et al., 2013). We therefore suggest that the ratio of coarse particles was an indirect response to the temperature rise





over the southern tropical Andes. A higher FLH increases the glacier terminus elevation, which reduces the glacial area and exposes sediments; these conditions facilitate the transport of coarse material to the ice core sampling site. These observations are similar to those previously observed in the European Alps (Oerlemans et al., 2009). However, the dust sources of the coarse

particles cannot be accurately determined due to the absence of high-resolution provenance studies.

Overall, we identified three phases of CPPn increase (Fig. 7). The first phase during 1947–1985 was characterized by a moderate retreat of the Illimani glaciers driven by an increase in the FLH, which promoted the rise in CPPn. The strong El Niño events during the second phase at 1985–1999 reduced snow precipitation and increased atmospheric temperatures, which accelerated glacial retreat and increased CPPn. The final phase since 1999 was characterized by reduced glacial coverage and

continued glacial shrinking in response to higher temperatures and periods of reduced snowfall driven by extreme conditions over the tropical Pacific and tropical North Atlantic. The recently deglaciated area became a new source of dust particles to the Illimani summit, which was the likely cause for the unprecedented rise in the coarse particle proportion during the 1990s.

## 4   Conclusions

The annually resolved mean dust concentrations (2–20 μm particle size range) in Illimani during 1919–2016 was ∼13,000 particles per mL. We identified two common periods of enhanced dust deposition when comparing the Illimani and Quelccaya ice core records (Thompson et al., 2013): 1) 1930s–1940s and 2) mid-1980s–2016. These enhanced dust periods coincided with drier conditions inferred from the observed minima in reconstructed snow accumulation rates and moisture transport from the Amazon basin.

The relative abundance of coarse dust particles in the Illimani glacier (expressed as the grain size index CPPn) correlated with SST in the tropical North Atlantic (TNA index). This suggests that a warmer tropical North Atlantic reduces the moisture advection from the Amazon basin to the Bolivian glaciers; this enhances drier conditions over the Bolivian Altiplano and promotes negative glacier mass balances, which mobilizes coarse dust particles from periglacial dust sources.

Glacier n°8 remained relatively stable from 1956 to 1983, but its area decreased by 17% in 2009. This coincided with an in-

crease in CPPn since the late 1980s. Furthermore, the CPPn trend correlated with a rapid increase in FLHs across the region (r = 0.41). Thus, increasing temperatures caused glacial shrinkage, which exposed more sediments and increased the deposition of locally sourced dust to the Illimani summit. We identified distinct phases in this mechanism since the 1950s. The first phase occurred until the mid-1980s and was characterized by moderate glacial retreat and a moderate increase in CPPn. As temperatures continued to increase from the 1980s to the 1990s, strong El Niño events and warmer conditions over the tropical North

Atlantic caused an unprecedented deposition of locally sourced dust particles in the 97-year ice core Illimani record.



*Data availability.* Data will be available at the NOAA (US National Oceanic and Atmospheric Administration) data center for paleoclimate after the acceptance of the paper: http://www.ncdn.noaa.gov/data-access/paleoclimatology-data/datasets/ice-core.

*Author contributions.* FL, JS, RR, PG, BD and GB wrote the manuscript; JS designed the research; FL, JS, PG, BD and GB conducted dust
analyses. RR and ER analyzed aerial photographs. PG, SK and VM provided analytical resources. All authors were involved in editing the manuscript.

*Competing interests.* The authors declare that they have no conflict of interest

*Acknowledgements.* The 1999 drilling campaign at Nevado Illimani was organized and covered by a joint French (IGE and IRD) and Swiss (PSI and University of Bern) project. The drilling campaign in 2017 was organized in the frame of the ICE MEMORY project (www.ice-
memory.org) supported by IRD, CNRS, UGA Foundation and local Bolivian institutions (Universidad Mayor San Andres – La Paz). Drilling equipment was provided by IGE and CNRS/INSU/C2FN CLIMCOR project (ANR-11-EQPX-0009-CLIMCOR). We thanks all participants taking part of the field campaigns. This study is part of the investigations of the Brazilian National Institute of Science and Technology of the Cryosphere (Brazilian National Council for Scientific and Technological Development - CNPq Process 465680/2014). FGL Lindau thanks CNPq (Processes 141013/2015-0 and 200496/2017-4) for scholarship to study at the EuroCold (University of Milano-Bicocca, Italy). We
thank JR Petit (IGE–CNRS, Grenoble) for helping in the 1999 dust analyses on the Illimani Ice core.



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





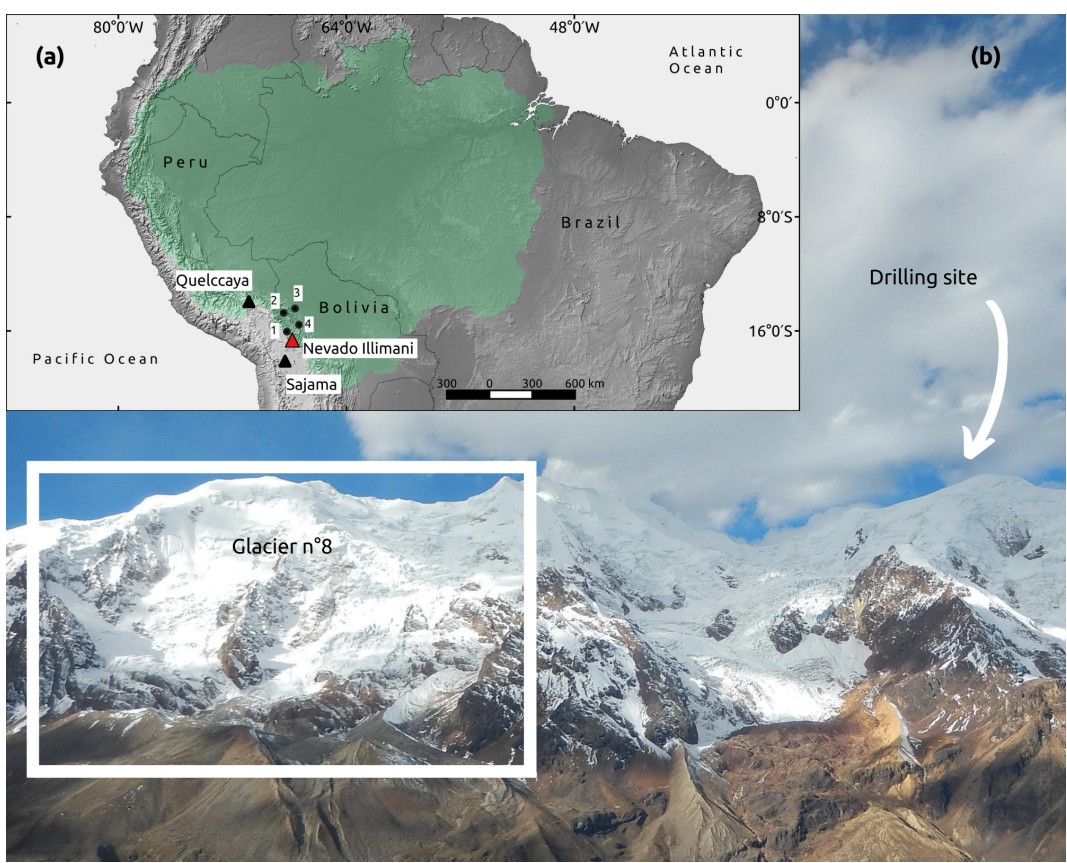

**Figure 1.** Map of the study region (a) indicating the locations of the ice core sites: Nevado Illimani (red triangle), Quelccaya (black triangle), and Sajama (black triangle). The green area delimits the Amazon basin (Mayorga et al. (2012); obtained from http://daac.ornl.gov). Also indicated are the Zongo glacier (1), and the meteorological stations in the Bolivian Amazon basin: Apolo (2), Rurrenabaque (3) and Sapecho (4). The land basemap was obtained from Natural Earth (http://www.naturalearthdata.com). (b) A photograph taken in June 2017 of the west/south face of Nevado Illimani, indicating the drilling site (snow and ice samples) and the selected glacier (n°8) that was analyzed by aerial photographs.



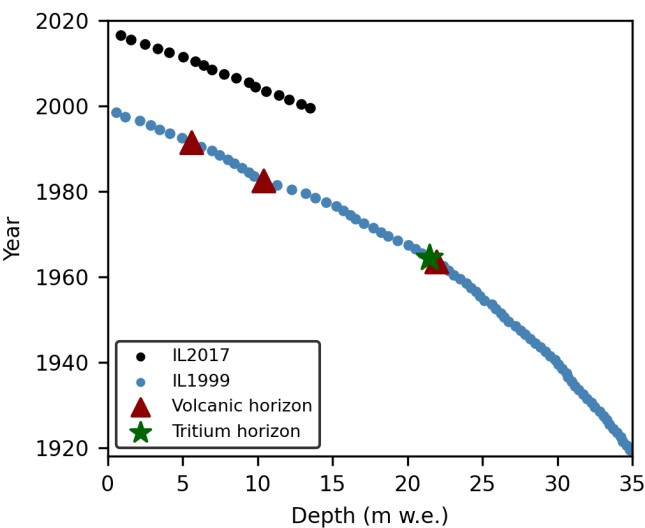

**Figure 2.** Depth age profiles for the IL1999 (blue points) ice core and the IL2017 (black points) firn core. The reference horizons in IL1999 are indicated by red triangles (1991, 1982, and 1963 volcanic layers) and the green star (1964 tritium peak).

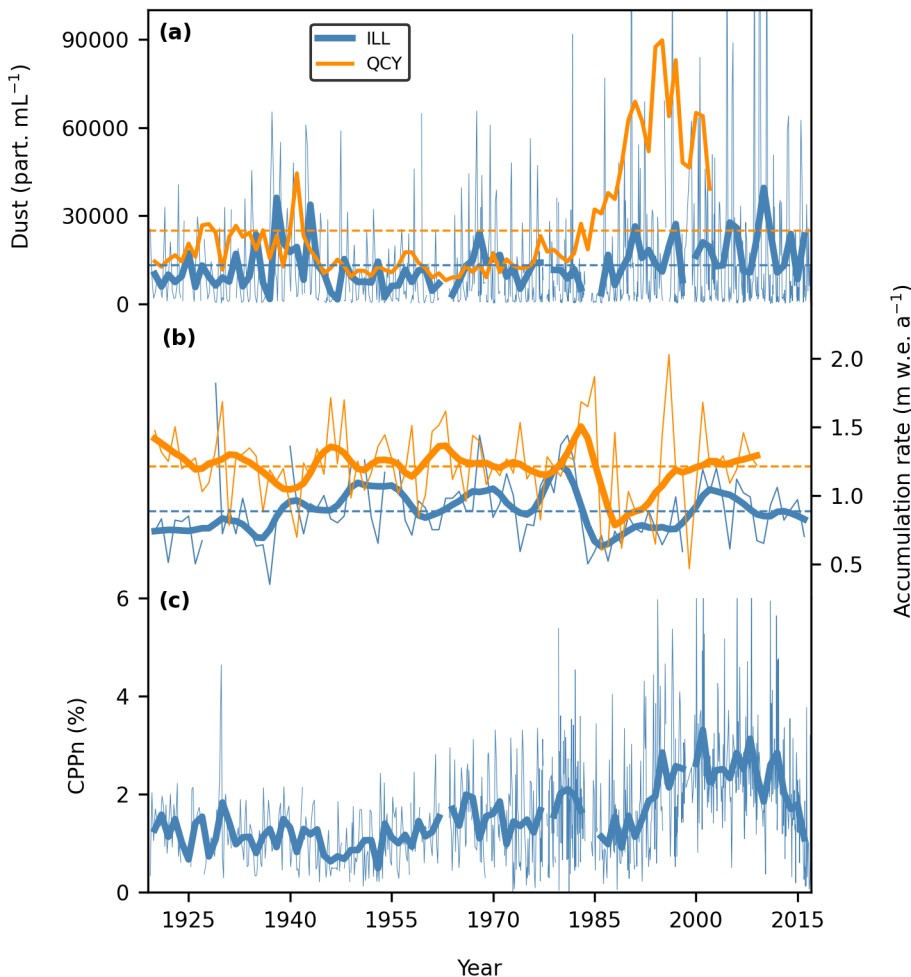

**Figure 3.** Comparison between dust concentration (a) and annual snow accumulation rates (b) in ice cores from Nevado Illimani (ILL, blue line, this study) and Quelccaya ice cap (QCY, orange line) (Thompson et al., 2013). Raw (thinner line) and annual (thicker line) dust concentrations from Illimani considers a particle size range of 2–20 μm. Conversely, annually resolved dust concentration from Quelccaya considers a particle size range of 0.63–20 μm. The thicker lines in (b) correspond to the 10-year LOWESS smoothed data, and the thinner lines refer to the raw data. The horizontal dashed lines refer to the mean annual dust concentration in (a) and the mean accumulation in (b) at each site. The raw (thinner line) and annual (thicker line) values in (c) refer to the coarse particle proportion in terms of number (CPPn) in the Illimani ice cores.

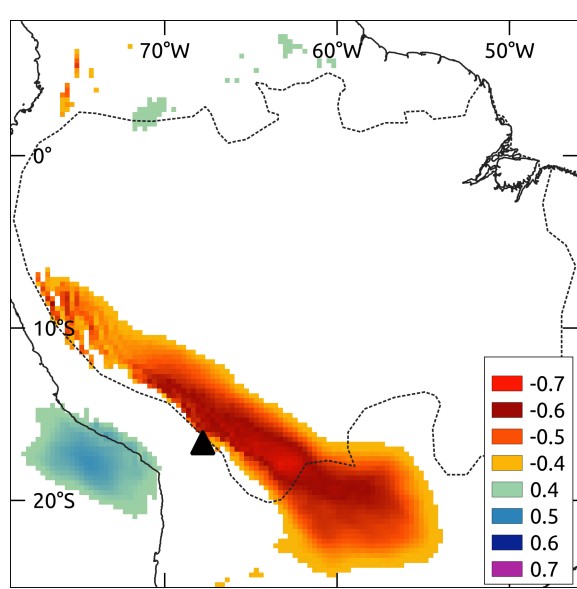

**Figure 4.** Spatial correlations (significant at the 95% level) between the 3-year moving average record of coarse particles (CPPn) and the specific humidity at the 500 hPa level (ERA5 reanalysis comprising the period 1980–2015). The black triangle locates the Nevado Illimani, and the dotted line delimits the Amazon basin.

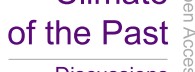

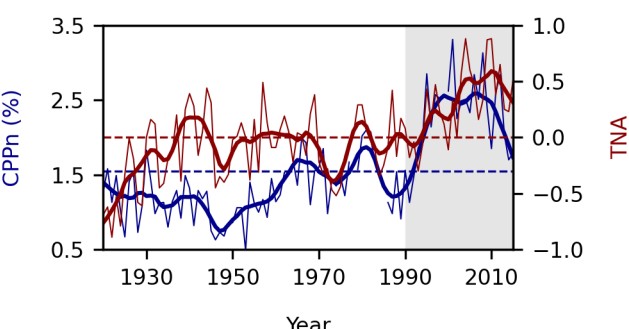

**Figure 5.** Relationship between the coarse particle percentage in terms of number (CPPn) and sea surface temperature anomalies in the tropical North Atlantic (TNA) referenced to the period 1981–2010. Thinner lines correspond to annually resolved data, and thicker lines correspond to the 10-year LOWESS smoothed data. Horizontal dashed lines highlight the mean for each parameter. The gray area indicates the rise of both CPPn and TNA during the 1990s.



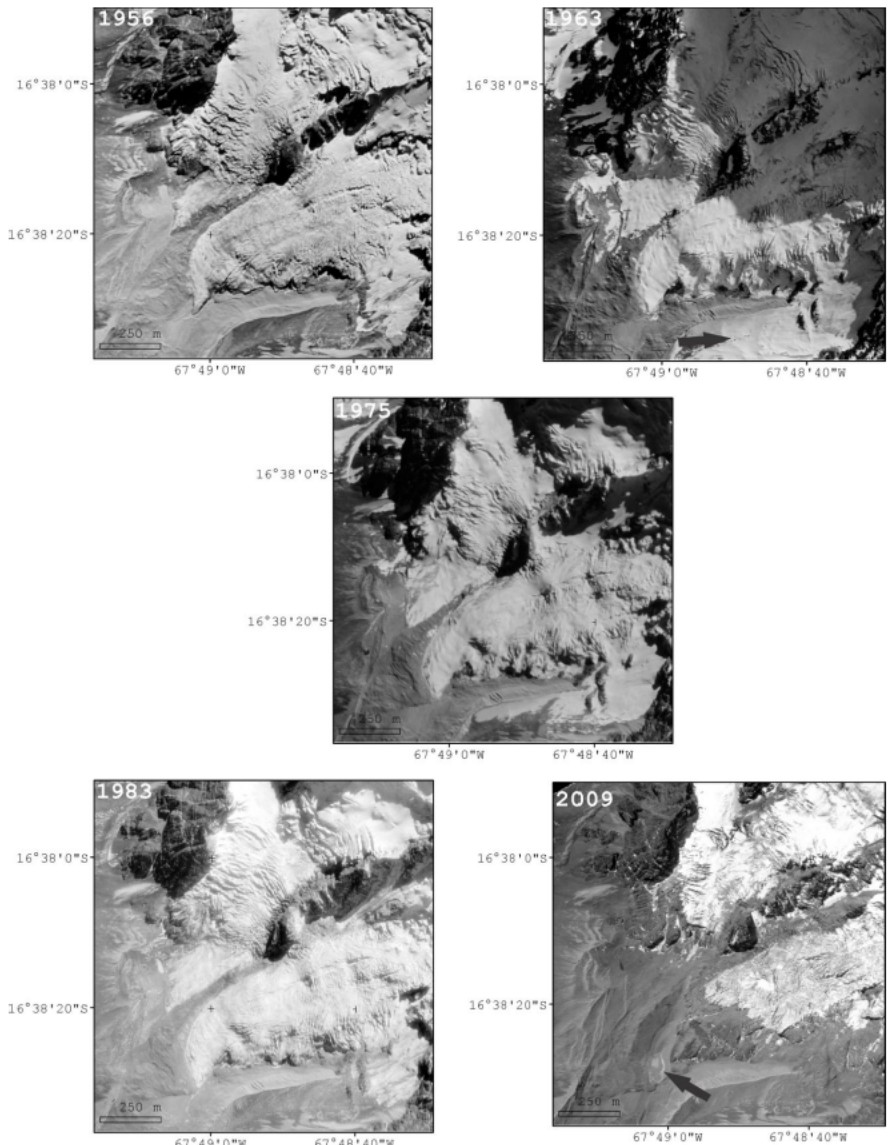

**Figure 6.** Aerial photograph sequences obtained for the analysis of glacier n°8 located in the southern sector of Nevado Illimani. They are based on twelve photographs, available from the Bolivian Servicio Nacional de Aerofotogrametria, digitized using a photogrammetric scanner. The arrow over the left sector of the glacier in 1963 points to recent snowfall, which masks the glacier's limit. The glacier shrinkage from 1983 to 2009 formed a small proglacial lake.





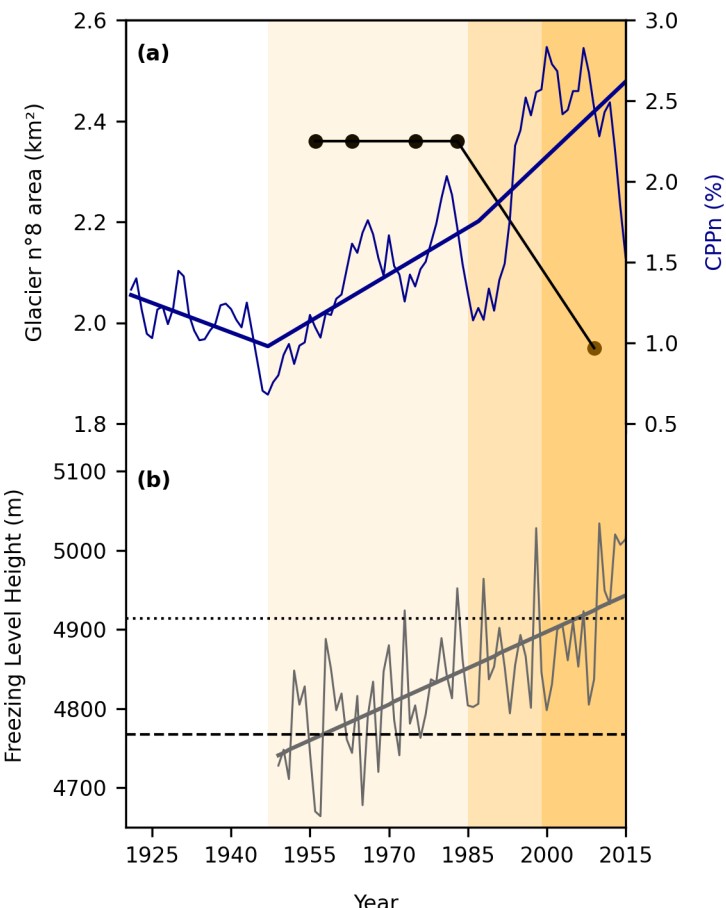

**Figure 7.** Relationships between glacial retreat, coarse particle percentage (CPPn), and the freezing level height over Nevado Illimani. (a) the variability in glacial area of n°8 (black line and black dots) compared with the annually resolved CPPn records (thinner blue line, 3-year moving average). The thicker blue line represents the linear piecewise fit applied to CPPn. (b) Annually resolved freezing level heights (gray line). The thicker line represents the linear increasing trend ($r^2 = 0.5$). The horizontal dashed and dotted black lines show the mean elevation of the glacial terminus for glaciers larger than $0.5 \ \mathrm{km}^2$ at Nevado Illimani in 1963 and 2009, respectively (Ribeiro et al., 2013). The orange bands highlight the three periods in the record (discussed in the text): 1950–1985 (light orange), 1985–1999 (orange), and 1999–2015 (dark orange).