# Peer review of "Dust record in an ice core from tropical Andes (Nevado Illimani – Bolivia), potential for climate variability analyses in the Amazon basin"

_Climate of the Past, 2020_

## Referee Comment (RC1) · Anonymous Referee #1 · 21 Oct 2020

This paper provides a nicely balanced combination of ice core, reanalysis data, remote sensing data and reference to previous work. Some suggestions follow: (1) The dust record comparison (Quelccaya vs Illimani) in Figure 3 is not entirely consistent with the text in the conclusion section. The conclusions refer to coincident dust increases in the 1930s-1940s and mid-1980s-2016. The more recent increase is not apparent in the Illimani record. More explanation is required in the text to address this issue. (2) The spatial correlation depicted in Figure 4 is based on a 3-year moving-average for the overlap period (Quelccaya and Illimani) of 1980-2002. The correlation is moderately

significant for this sampling resolution. What is it for an annual match realizing there is a 1-2 year dating error? If only the 3 year correlation is shown and it only covers 1980-2002 this is a match based only on 14 data points. (3) For Figure 5 the 10 year smoothed correlation looks similar but the correlation for the annual data looks inverse? Please explain more in the text.

---

## Referee Comment (RC2) · Anonymous Referee #2 · 18 Nov 2020

In this manuscript, Gaudie and colleagues investigate the link between dust deposition on the Illimani glacier in Bolivia, and climate change and its associated glacier retreat in the past 100 years. They improve on previous studies that measured dust concentrations by also analyzing size distribution. My main criticism is that the authors base all their conclusions on the correlation of the coarse dust particle fraction with various other variables. Science is not about finding correlations between variables, but about explaining dynamical links when correlation occurs. This manuscript lacks source attribution, trajectory modeling, and other supporting evidence that would substantiate

the conclusions. In addition, the scientific novelty is poor, as many of the conclusions could be reached (and were reached in other studies) without the dust contribution. For these reasons I suggest to reject this manuscript at this point. I would recommend to the authors to split the dust and satellite sections into two separate papers and improve these by adding more substantial analyses than just correlation studies.

Major comments:

The authors split the size distribution into a CPP (10-20um) fraction and argue that local dust sources will increase the CPPn fraction. This could be the case, but it is not shown. As a counter example, one could have a regional source with mode around 10um and a local source with a mode around 40 um. In that case, all CPPn variability would be due to emission and transport changes for the regional source. I suggest to show the whole range of size distribution measured by the Multisizer 4 up to 60 um and discuss potential local and regional sources based on that total distribution.

Figure 4: I put these remarks in the major comments as the interpretation of this figure is central for the manuscript. In line 154-155 the authors say that the correlation between CPPn and the specific humidity "suggests lower regional precipitation and reduced moisture transport from the Amazon basin to the southern tropical Andes". That is a very strange way to interpret a correlation. Reduced regional precipitation could easily be shown using the ERA5 data instead of that correlation plot. In addition, there is no justification for the reasoning behind the reduced moisture transport from the Amazon. It could just as well be that both CPPn and specific humidity react similarly to temperature changes. A simple correlation between these two variables is not enough to justify that causal link. Finally, I will note that the paper by Segura et al. that is cited at the end of that sentence only talks about DJF precipitation, while the precipitation over the study cite is "distributed over 9 months" according to line 45.

In line 182 the authors state "From 1983 to 2009, we estimated a height reduction of $-13$ m and a negative mass balance of $-11.8 \pm 2.29$ m w.e.". There is no information

in the methods about how they estimated that. In addition, this simple phrase thrown casually here should be a paper by itself, with lines 186-194 as part of its introduction.

The authors link the regional warming trends with rising CPPn fractions, arguing that glaciers melted during periods with warmer temperatures, which produced a greater abundance of large particles. However, the CPPn fraction is decreasing during the first warming period established by Gilbert et al., 2010 (1920-1960). This contradicts the interpretation of the authors and is not discussed in the manuscript.

Minor comments:

Line 40: The link between measured dust concentration variability in glacier ice cores and changes in dust aerosol concentration in the air is complex. Please rephrase more conservatively.

Chapter 2.3: The exponential fit regression may give a good first order approximation of the accumulation rate. However, I think the authors should show that sublimation and surface melt are secondary processes and should not affect the accumulation rate estimate significantly.

Figure 1: The location of Glacier 8 and of the drilling site are not clear.

Line 139: Why is the mean only taken from the period 1999-2016 when the figure shows data from 1920?

Lines 155-156: Figure 4 does not "indicates reduced deep convection over the Bolivian Amazon". It's a correlation plot, nothing more.

Figure 5: The caption says the think lines represent 10-year LOWESS smoothed data, while in the text a correlation of 3-year moving average of CPPn with TNA is consistently mentioned. Choose one and use it for both the correlation and the figure. You probably want to correlate CPPn and TNA sst after smoothing both, not just the CPPn.

Line 170: TNA SST was positively correlated...

Lines 172-173: Why the Nino 4 region? Air parcels arriving to the Bolivian Plateau are more likely to have originated from Nino 1 or 2 regions.

Line 181: What does it mean that the DEMs have high mean absolute errors? Please expand.

Lines 197-202: It looks like the sections for the piecewise linear function were subjectively chosen to suite the interpretation of the authors. A different choice of sections would lead to a completely different interpretation. Please use a more robust and objective method.

Line 207: Could be. Or not. This statement needs some supporting evidence.

---

## Referee Comment (RC3) · Anonymous Referee #3 · 7 Dec 2020

This paper seeks to investigate the relationship between coarse dust particles and climate variability. In essence, the authors argue that periods of stronger glacier retreat lead to increased exposure of glacial deposits that provide for increased coarse particle deposition on the ice. The paper is interesting and could eventually make a significant contribution to our understanding of the dust and its climatic significance in tropical ice cores. There are, however, several aspects in the applied methodology that are flawed and need to be changed to make sure that their reported results are indeed robust. I have outlined my major concerns and a few smaller suggested edits below.

[Figure]

Major concerns

1) I am concerned about the way the correlation with the SST field was performed. This analysis should not be based on annual means (calendar years). Doing this will lead to biased results as it cuts the Illimani wet season artificially into two separate years. Similarly, ENSO is phase-locked to the seasonal cycle and you will lose the strong ENSO signal if the data are averaged over calendar years. The correlations with SST need to be performed separately for wet and dry season or at the very least you need to define a hydrologic year (e.g. July-June).

2) Another problem is that apparently time series were not detrended prior to correlation analysis. Many of the records used, are characterized by strong trends, which must be removed prior to such an analysis. The SST in the tropical N. Atlantic and in the Nino 4 region, for example, have warmed significantly over the 2nd half of the 20th century (Fig. 5 and Fig. S3). The CPPn record also shows a strong upward trend. Hence you will obtain a significant correlation between these records, simply because they show similar long-term trends. But these are spurious correlations that are simply an artifact of the trends themselves; they do not imply that the records are casually connected or significantly correlated on a yearly basis. This analysis therefore needs to be performed on detrended series. The same comment also applies when correlating CPPn with precipitation records from nearby stations, with 500 hPa specific humidity (Fig. 4) or with the FLH (Fig. 7). If there is indeed a mechanistic link between all these records, the correlations will still be significant after removal of the trends.

3) For some variables, a 3-yr running mean filter was applied prior to calculating the correlation coefficient. How was the resulting reduction in the degrees of freedom taken into account? This should be discussed in the methods section.

4) The choice of the Nino 4 index to represent the Pacific connection with Illimani requires better justification. As shown by Francou et al. (2003), the mass balance in the Cordillera Real is much more closely related SSTA in the eastern Pacific (Nino1+2

or Nino3 region).

4) The discussion of the FLH changes are interesting. But I would encourage the authors to compare the results to other recent studies that have also looked at FLH changes (and related ELA changes) nearby, notably on Quelccaya (Yarleque et al., 2018) and on Zongo glacier (Vuille et al., 2018). This would provide for additional confirmation, as the FLH is a large scale tropospheric change that should be consistent across all three sites, but varies depending on the reanalysis product used.

5) More rigor is required in the analysis of Figures 6. It is not sufficient to simply show a sequence of aerial photographs. Since you digitally analyzed their extent in each image, this change from one time period to the next should be highlighted in color contours to make the changes visible.

6) In addition, the methodology to delineate the glacial extent needs to be discussed in more detail and an error assessment needs to be performed, considering uncertainties associated with the hand digitizing of the glacier extent, the spatial resolution of the digitized aerial photographs and the resolution and vertical accuracy of the underlying DEM. The glacier extent in Figure 7 cannot be indicated by a single point, but needs to include error bars.

7) The retreat of glacier 8 in Figure 7 is pinned on just one data point in 2009. It would be helpful to have a few more data points between 1983 and 2009 to reduce the uncertainty in this response.

8) Figure caption 7: You state "The thicker line represents the linear increasing trend (r2 = 0.5)." What does that mean? Trends are not characterized by coefficients of determination (r^2), but by their slope and testing whether the slope is significantly different from zero (e.g. with an F-test).

9) In the discussion of the relationship between temperature on Illimani and glacier retreat, the argument is made that they tend to coincide. This may indeed be the case,

but in general glacier retreat is considered a delayed response, which, depending on glacier size and dynamics, may not express itself until a decade or two after the warming started. A bit more discussion on the anticipated response time of this particular glacier to such forcings would be helpful in this context.

Minor issues:

Line 17 and throughout entire paper: The Andes are usually considered to be plural, hence it should be 'The tropical Andes have'.

Line 61: down to bedrock

Line 98: from 1941

Line 141 replace 'finishes' with 'ends'

Line 173-174: You write that 'Tropical Pacific SST anomalies are associated with anomalous upper-level westerlies over the southern tropical Andes'. Note that this applies only to periods with anomalously warm tropical Pacific SST anomalies (e.g. see (Garreaud et al. 2003).

References cited in review

Francou, B., et al. 2003: Tropical climate change recorded by a glacier in the central Andes during the last decades of the twentieth century: Chacaltaya, Bolivia, 16°S. J. Geophys. Res., 108, D5, 4154, doi: 10.1029/2002JD002959.

Garreaud, R., et al. 2003: The climate of the Altiplano: Observed current conditions and mechanisms of past changes. Palaeogeogr. Palaeoclimatol. Palaeoecol., 194, 5-22.

Vuille, M., et al. 2018: Rapid decline of snow and ice in the tropical Andes – Impacts, uncertainties and challenges ahead. Earth Sci. Rev., 176, 195-213.

Yarleque, C., et al. 2018: Projections of the future disappearance of the Quelccaya Ice

[Figure]

Cap in the Central Andes. Sci. Rep. 8:15565, doi:10.1038/s41598-018-33698-z.